# Kinesthetic motor-imagery training improves performance on lexical-semantic access

**Camille Bonnet**[1,2☉], **Mariam Bayram**[1☉], **Samuel El Bouzaïdi Tiali**[1], **Florent Lebon**[3], **Sylvain Harquel**[1,4], **Richard Palluel-Germain**[1], **Marcela Perrone-Bertolotti**[1,5]*

**1** Univ. Grenoble Alpes, Univ. Savoie Mont Blanc, CNRS, LPNC, Grenoble, France, **2** Psychological Sciences Research Institute, Université Catholique de Louvain, Louvain-la-Neuve, Belgium, **3** INSERM UMR1093-CAPS, Université Bourgogne Franche-Comté, UFR des Sciences du Sport, Dijon, France, **4** Defitech Chair of Clinical Neuroengineering, Center for Neuroprosthetics (CNP) and Brain Mind Institute (BMI), Swiss Federal Institute of Technology Lausanne (EPFL), Campus Biotech, Geneva, Switzerland, **5** Institut Universitaire de France, Paris, France

☉ These authors contributed equally to this work.

* marcela.perrone-bertolotti@univ-grenoble-alpes.fr

**Data Availability Statement:** All data, analysis code, and research materials are available on the Open Science Framework (https://osf.io/bw598/).

## Abstract

The objective of this study was to evaluate the effect of Motor Imagery (MI) training on language comprehension. In line with literature suggesting an intimate relationship between the language and the motor system, we proposed that a MI-training could improve language comprehension by facilitating lexico-semantic access. In two experiments, participants were assigned to a kinesthetic motor-imagery training (KMI) group, in which they had to imagine making upper-limb movements, or to a static visual imagery training (SVI) group, in which they had to mentally visualize pictures of landscapes. Differential impacts of both training protocols on two different language comprehension tasks (i.e., semantic categorization and sentence-picture matching task) were investigated. Experiment 1 showed that KMI training can induce better performance (shorter reaction times) than SVI training for the two language comprehension tasks, thus suggesting that a KMI-based motor activation can facilitate lexico-semantic access after only one training session. Experiment 2 aimed at replicating these results using a pre/post-training language assessment and a longer training period (four training sessions spread over four days). Although the improvement magnitude between pre- and post-training sessions was greater in the KMI group than in the SVI one on the semantic categorization task, the sentence-picture matching task tended to provide an opposite pattern of results. Overall, this series of experiments highlights for the first time that motor imagery can contribute to the improvement of lexical-semantic processing and could open new avenues on rehabilitation methods for language deficits.

## Introduction

Multiple lines of evidence show that the motor system is recruited during language processing. More specifically, access to action-related concepts is thought to rely, at least partially, on distributed neural networks that encompass the sensorimotor system [1, 2]. To illustrate,

**Funding:** The author(s) received no specific funding for this work.

**Competing interests:** The authors have declared that no competing interests exist.

neuroimaging studies have revealed that action-related language processing elicits activation in motor and premotor brain regions [1–7], in a somatotopic manner [1]. For instance, reading or listening to verbs like "kick" or "grasp" induces leg and hand sensorimotor activation, respectively. Consistent with the embodied cognition framework, authors have hypothesized that these motor activations might facilitate action language processing. Results from studies using Transcranial Magnetic Stimulation (TMS) have supported this hypothesis by showing that TMS applied to the leg and arm motor area, while participants are asked to perform a lexical decision on leg-related and arm-related action verbs, specifically reduces the processing time when the stimulation site matches the verb's effector [8]. On the contrary, inhibitory stimulation induced by repetitive-TMS has been shown to increase the processing time of the actions associated with the represented effector [9]. Additional evidence has been provided by electrophysiological studies indicating that these motor activations arise as early as 200 ms following verb presentation, suggesting that the motor system might play a facilitatory role in lexical-semantic processing [1, 2, 4, 9]. However, results from behavioral experiments are somewhat less straightforward. Indeed, although the action-sentence compatibility effect is well documented in the literature, with results showing a facilitation of action-language processing in congruent language-effector contexts [4, 10–12], some studies reported an opposite pattern of results, therefore pointing towards an interference effect [13, 14].

The implication of the motor cortex in concrete action-verb processing is frequently compared to its implication in the processing of abstract verbs (i.e., without a specific action representation, such as "to accept"). Although the role of motor activation in abstract-concept processing remains to be defined, studies have shown that the processing of abstract words is likely to involve motor regions, yet to a lesser extent than concrete language. For instance, when asked to read sentences varying in their level of concreteness (i.e., from purely concrete to merely abstract), results from the fMRI study of Sakreida and colleagues [15] indicated that all sentences elicited the activation of core areas of the sensorimotor neural network, independently of their level of concreteness. However, the magnitude of these activation was greater for concrete sentences and specific activation in the language processing system were additionally found for abstract sentences ([15]; for a review, see [16]). A similar pattern of results was documented in TMS studies. For instances, Reilly and colleagues [17] showed that the stimulation of the primary motor cortex (M1) at 300 ms after the verb presentation impairs the comprehension of literal and metaphoric action (i.e., conveying abstract language) sentences, suggesting that M1 is involved in abstract-sentence processing. Hence, results from neuroimaging studies suggest that both abstract and concrete language processing elicit sensorimotor areas' activation. However, a greater activation of motor areas in response to concrete language processing has been put forward, which is corroborated by behavioral studies indicating that concrete language processing is faster than abstract processing [18, 19].

Several studies conducted in clinical populations supported the previously suggested language and action relationship, and more specifically results showing that changes in the motor system can affect the processing of action concepts. For instance, Bocanegra et al. [20] reported the presence of a selective impairment in the recognition of high-motion action verbs (e.g., "to run") compared to low-motion action verbs (e.g. "to sleep") in patients with Parkinson's disease (without mild cognitive impairment). Interestingly, similar results were observed in healthy participants who underwent temporary motor deprivation: Bidet-Ildei and colleagues [21] showed that a 24-hour upper limb immobilization induced a significant reduction in performance on hand-action verb categorization compared to foot-action verb categorization tasks.

These results suggest that action word processing may partially depend on the motor system, which is consistent with the embodied view of language. This view suggested that brain

structures traditionally associated with perceptual and motor processes are also involved during action concepts processed in verbal modality [22–24] and could be explained by the existence of a mechanism that re-enacts or simulates the motor experience evoked by action concepts [24–27]. This simulation mechanism implies the generation of an action plan as well as the prediction of the action's sensory consequences, as previously proposed for instance, in covert verbal production [28, 29]. Consequently, it is proposed that the early activation of motor areas elicited during action concept processing may reflect a motor simulation [25, 26, 30–34]. Based on this assumption, some authors postulated that motor system training can lead to a functional improvement in language comprehension and production, by reinforcing the functional relationship between the two systems [34–36].

In line with this prediction, Beilock et al. [35] showed that participants with specific sensori-motor expertise (i.e., ice hockey professional players and fans, who hold visuomotor and visual expertise, respectively) exhibited better comprehension of sentences semantically related to such domain of expertise (e.g., "The hockey player finished the shot") compared to unrelated sentences (e.g., "The individual pushed the cart"). This improvement in behavioral performance was mediated by the level of activity in the left dorsolateral premotor cortex (i.e., an area usually involved in action selection and implementation). The authors suggested that sports' experience may modify the neural networks that underlie language comprehension by consolidating their connections to motor brain regions implicated in sports' performance [35, 37]. If sensorimotor expertise—acquired through action observation and action execution [35, 37]—can modify the motor system and its connections to the language system, we assume that motor imagery (i.e., the mental simulation of an action without concomitant movement production, [38] may also induce changes in the motor system and its functional connectivity. Indeed, the existence of an overlap in brain activation during motor imagery (MI), actual action execution and action observation [39] have been well established, suggesting that these different processes share similar motor representations [27, 40]. Specifically, neuroimaging studies revealed that imagined and executed movements share common underlying neural networks, including structures such as the basal ganglia, cerebellum, primary motor and premotor cortices (particularly the supplementary motor area), inferior parietal cortex, prefrontal cortex [38, 39, 41–43].

In the domain of voluntary movement, MI practice has been suggested to be an efficient technique for motor learning in healthy individuals (e.g., athletes, musicians, for a meta-analysis, see [44], and as a motor rehabilitation technique for patients with various pathologies (see [45–47] for stroke patients; [48] for spinal cord injury; or [49] for Parkinson's disease). MI is known to enhance corticospinal excitability and engage motor regions that are active during action-language processing. Hence, MI could elicit neuroplastic adaptations in the cortical representation of movement [50, 51]. Moreover, evidence in the literature indicates that repeatedly executing or imagining an action can affect language proficiency in patients with aphasia [45, 52–55]. Interestingly, Chen and collaborators [52] exploited this connection between the language and sensorimotor system and developed an action-observation training known as the Intensive Language-Action Training (ILAT). They observed that this training improved language functions to a comparable extent as observed with conventional speech therapy programs. Interestingly, they also observed that ILAT-based neuroimaging results were related to stronger activation in brain areas involved in speech comprehension (e.g. the superior temporal gyrus), speech production (e.g. the inferior temporal gyrus), and brain regions involved in postures and gestures identification (e.g. the supramarginal gyrus) as compared to a dynamic-object observation training.

In conclusion, several studies highlighted that MI increases corticospinal excitability and engages motor regions that are involved during action-language processing. In addition,

studies evaluating the neural correlates of MI, showed that MI, action execution, and observation involve a common brain network. They also showed that action execution and observation induce improvement in language comprehension [35, 37]. Given the arguments presented above, we hypothesized that imagining the execution of movements might also improve language processing. From a clinical perspective, and compared with action execution, MI-based training has the advantage that it can be used in populations with limited motor functioning, and requires little-to-no physical exertion. The objective of the following experiments was to investigate whether MI practice might induce a facilitatory effect on lexical-semantic processing in healthy subjects. For this purpose, participants were included in two different training programs: an experimental kinesthetic MI training group or a control static visual imagery training group. Training effects were evaluated in two different language comprehension tasks. In the first experiment, a single training session was proposed, and between-group differences were examined. In a second experiment, participants completed several training sessions and their performance on language comprehension tasks were assessed pre- and post-training.

## Experiment 1

This experiment aimed to evaluate the impact of a single kinesthetic motor-imagery training session on language comprehension and was organized in three steps. The first step (Step 1) assessed pre-training imagery abilities. In the second step (Step 2), participants were included in one of the two training group and received a specific training session according to the group they belonged to: 1) an experimental group—kinesthetic motor-imagery (KMI group) —in which participants were instructed to imagine themselves executing a movement and the specific sensations elicited by the imagination of the said movement and 2) a control group— static visual imagery training (SVI group)—in which participants were instructed to imagine several landscapes as if they were looking at a picture "in their mind" (see Step 2: training, for details). The SVI training condition served as the control condition, as participants in this condition received an active imagery training without any motor component. We hypothesized that visual imagery would not enhance language comprehension and would not implicate the motor system. After the training session, all participants were evaluated on two semantic tasks (Step 3): i) a semantic categorization task, in which participants were instructed to judge as quickly as possible whether an auditory presented verb was concrete or abstract. ii) a sentence-picture matching task, in which participants were instructed to indicate the picture, among two alternatives, that best matched a sentence auditory presented in white noise. The two tasks included action verbs (e.g., to grasp) and abstract verbs (e.g., to think). Some of the action verbs referred to previously imagined actions by participants in the KMI group (i.e., target verbs). Following the embodied view of language, which has proposed that language is grounded in action, we hypothesized that if a kinesthetic motor imagery training can indeed improve lexical-semantic access, better performance on both language tasks are to be expected from participants in the KMI group than from those in the SVI control group (given that both tasks are driven by lexical-semantic processing and even though the sentence-picture matching task requires an evaluation of visual-semantic similarities [19, 56]. In addition, if the proposed KMI training protocol can induce specific effects for the trained imagined movements only (i.e., simple motor-to-semantic priming), we expect to find greater differences between concrete and abstract verbs, and target versus non-target items in the KMI group compared to the SVI control group. However, in case the KMI training protocol can induce training transfer effects, the magnitude of the difference between concrete and abstract verbs in one hand, and target versus non-target items in the other hand might be substantially reduced in the KMI

group. The hypothesized underlying mechanisms rely in a spread of motor activations (i.e., most likely primary motor areas) along with neuroplasticity effects if the training expand to non-target items only; or larger spread of activations to sensorimotor areas (i.e., most likely higher-order motor areas in addition to primary sensorimotor ones) along with neuroplasticity effects, if transfer effects are not only noticed on non-target items, but also on abstract ones.

## Method

All data, analyses' coding scripts, and research materials are available at: https://osf.io/bw598/?view_only=be42377bdb2e4b049399779b19d94bee

**Participants.** Thirty-six undergraduate students from the University Grenoble Alpes participated in this first experiment (33 women and 3 men). Participants were right-handed, native-French speakers with normal or corrected-to-normal vision. None of them reported having a history of neurological, psychiatric, or auditory pathologies. Following Step 1 of the protocol, participants were randomly assigned to two different groups: half of them were assigned to the experimental group and received kinesthetic motor imagery training (i.e., KMI group, M = 19.6 years; SD = 1.4 years) and the other half were assigned to a control group and received static visual imagery training (i.e., SVI group, M = 20.6 years; SD = 2.3 years). Participants received an explanation of the procedure and gave written informed consent. They were informed about the objectives of the project at the end of the experiment. The study was approved by the local ethical committee (CER Grenoble Alpes Avis-2018-12-11-2).

**Tasks and procedure.** To minimize the potential influence of the experimental demand characteristics [57], the cover story presented to the participants was that they were taking part in two distinct experiments: one project in which we aimed to assess a new language comprehension task in French and another project in which we aimed at assessing to what extent mental imagery abilities can be improved by training. Each experimental session lasted about one hour.

*Step 1*: *Pre-training assessment*. Participants filled in a survey containing questions about sport and music practice (i.e., some factors that could affect MI abilities [58] and completed the Kinesthetic and Visual Imagery Questionnaire (KVIQ-10; [59]. The 10-item KVIQ-10 was included as a measure of participants' ability to visualize and feel imagined movements. In this questionnaire, participants were asked to execute a movement once, then imagine themselves doing it while focusing on the intensity of the kinesthetic sensations elicited by the imagery (kinesthetic subscale), or on the visual clarity of the imagery (visual subscale). The movements that participants executed and then imagined were forward shoulder flexion, thumb-fingers opposition, forward trunk flexion, hip abduction, and foot tapping. On the visual subscale, they rated their imagery as "5 = Image as clear as seeing", "4 = Clear image", "3 = Moderately clear image", "2 = Blurred image", or "1 = No image". On the kinesthetic subscale, they rated their kinesthetic sensations as "5 = As intense as executing the action", "4 = Intense", "3 = Moderately intense", "2 = Mildly intense", or "1 = No sensation". Participants were also asked to perform a mental chronometry task [60], which measured the temporal correlation between real and imagined movements. This correlation is known to be an indicator of MI abilities [58], with a higher correlation indicating relatively better motor imagery abilities [60]. In the present study, participants were asked to imagine writing the address "2 Rue de la Libération" presented to them on a computer screen, before writing it down. Specifically, participants imagined writing the address upon hearing a sound and pressed a button to indicate they finished. This step was repeated for 5 trials, which provided an average duration estimation of the imagined movement. Afterwards, participants wrote down the address upon hearing the same sound and pressing the same button to indicate when they finished, for 5 trials as well. Average

reaction times across the 5 trials for the imagined and executed movements were used as individual scores [61]. For each group, the correlation between imagined and executed movements average durations was computed. Stimuli presentation and data collection were done using E-Prime (Psychology Software Tools Inc., Pittsburgh, USA).

*Step 2*: *Training*. After completing the first step, participants were randomly assigned either to the KMI-training group or to the SVI-training group. In the KMI training group, participants were instructed to imagine the execution and the specific sensations (e.g., muscle tension) related to the effector's movement. We chose the KMI modality as it activates the motor neural network to a greater extent compared to visual MI [62]. Participants had to imagine eight upper-limb movements: "to applaud", "to sweep", "to button up", "to write", "to slap", "to throw", "to paddle" and "to pour". Each session was conducted as follows: after the experimenter reminded instructions of the participants, a picture (previously associated with a movement during the familiarization phase) was projected on a white wall using a video projector during 2s. After the picture presentation, an auditory signal indicated that participants should close their eyes and start imagining the movement corresponding to the picture during 10s. After those 10s, a second auditory signal indicated them to open their eyes for a 10s-rest period. During this period, the participants had to evaluate the intensity of the sensation experienced while imagining the movement on a scale ranging from 1 to 5 (identical to the kinesthetic scale used in the KVIQ-10). This evaluation step allowed us to maintain participants active during the training session and to make sure they performed the task. A training session consisted of four training blocks, with each block including eight movements and a 1-minute rest period between each block. Pictures were randomly presented across the training block. The participants remained seated during the training session and had to remain silent, to prevent them from verbally describing any movement.

The SVI-training protocol had a similar structure to the one used in the KMI protocol. Participants were asked to mentally visualize eight landscapes as if they were looking at a picture "in their mind" [63]. After being reminded of the instructions, they were asked to perform four training blocks, each block including eight pictures (with a 1-min rest period between each block). Every landscape had to be mentally imagined for 10s with closed eyes. During the 10s rest-time, participants evaluated the visual clarity of their mental image on a scale ranging from 1 to 5 (identical to the visual scale used in the KVIQ-10). Each training (KMI and SVI) session lasted about 15 minutes.

*Step 3*: *Post-training assessment*. During this step, participants performed a semantic categorization task [12, 21] and a sentence-picture matching task [35, 37].

In the semantic categorization task (see Fig 1A), participants were instructed to categorize, as quickly and accurately as possible, the auditory-presented verbs as concrete (i.e., action verb, e.g., to draw) or abstract (i.e., state verb, e.g., to love) with a button press (see Fig 2A). The stimuli consisted of 56 French verbs selected from the French database Lexique.org [64] and divided into four categories: 14 target verbs (i.e., concrete upper-limb verbs that were related to an action presented during the MI training phase; e.g., to write); 14 non-target verbs (i.e., concrete verbs unrelated to the trained actions of the MI training phase, e.g., to caress); 14 abstract verbs (e.g., to admire); and 14 abstract fillers (allowing for an equivalent number of responses across the two categorization levels). Fillers were excluded from data processing. For the three verb categories (i.e., target, non-target, abstract), pair-by-pair comparisons were conducted and showed that the verbs did not differ on any of the following psycholinguistic variables: lexical frequency, number of homophones, number of letters, number of phonemes, number of orthographic neighbors, number of phonological neighbors, orthographic uniqueness point, phonologic uniqueness point, and number of syllables. Before the test phase, participants performed a 10-trial training phase with feedbacks (5 concrete verbs and 5 abstract

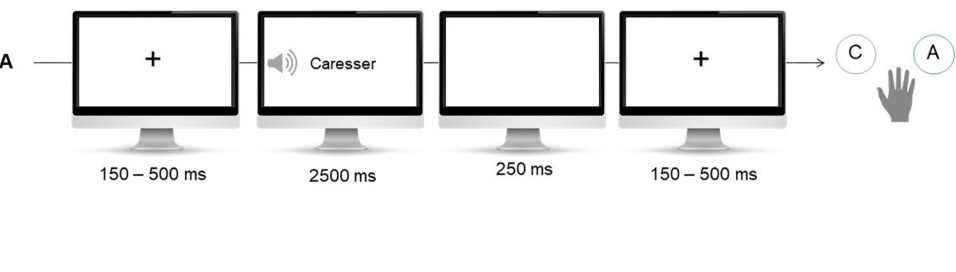

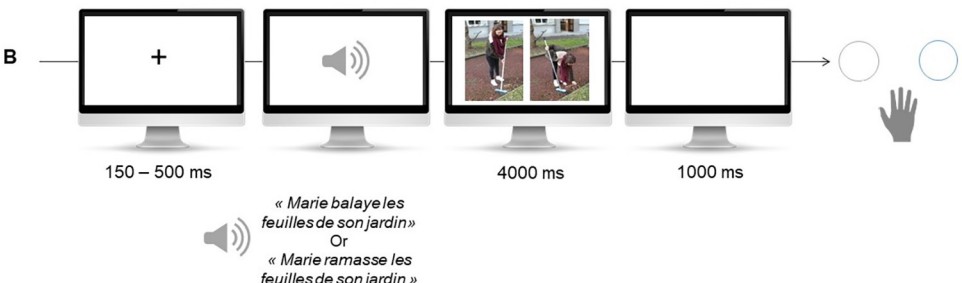

**Fig 1. Language comprehension tasks used.** Sample trials for the A) semantic categorization task and the B) sentence-picture matching task. *"Caresser"*: *"to caress"* (i.e., action non-target verb). *"Marie balaye les feuilles de son jardin"*, *"Marie is sweeping the leaves in her garden"* (i.e., matching condition with action target verb) or *"Marie ramasse les feuilles de son jardin"*, *"Marie collects leaves from her garden"* (i.e., mismatching condition). The individual in this picture has given written informed consent (as outlined in PLOS consent form).

verbs, different from those used in the experimental phase). Then, a fixation cross appeared for a duration varying between 150–500 ms, to avoid anticipation errors. Participants heard, through headphones, an isolated French verb in the infinitive form (e.g., "applaudir", "aimer"). After participants answered with a right-hand button press, or 2500 ms in absence of a response, a white screen appeared for 250 ms before the beginning of the next trial. The total number of stimuli presented during the test phase was 112 (the 42 experimental verbs were presented twice in two different blocks, such as the 14 abstract fillers, see Fig 1A). All stimuli were presented using E-Prime software (v2.0.10.353, E-Prime Psychology Software Tools Inc., Pittsburgh, USA). Manual responses were performed with the dominant right-hand and were made on a compatible E-prime SRBox. Accuracy and reaction time were recorded. Auditory stimuli were presented in a headphone with a comfortable sound level set by participants.

In the **sentence-picture matching task** (see Fig 1B), participants were instructed to judge as quickly as possible which picture, among two alternatives, best matched an auditory-presented sentence. This task was included as a more ecological measure of language processing, as the stimulus depicting actions was embedded in a realistic scenery. Before the test phase, participants performed 8 training trials with feedback (including 4 matching trials and 4 non-matching trials, see Fig 1B). Stimuli used during training were different from those used in the test phase. One typical trial included: a central fixation cross for 150–500 ms on screen followed by an auditory-presented French sentence (narrated by a female speaker) presented for 2000 ms. After the sentence's listening phase, two pictures appeared on the screen, both depicting an action performed by the same individual. The pictures remained on the screen until participants manual response (button press), or for a maximum of 4000 ms. Forty-two sentences were presented and were included in three different categories according to the type of verb used to build the sentence (i.e., using the same verbs as in the semantic categorization task). Fourteen sentences included a target verb (e.g., *"Marie balaye les feuilles dans son jardin"*, *"Marie is sweeping the leaves in her garden"*), 14 sentences included a non-target verb (e.g.,

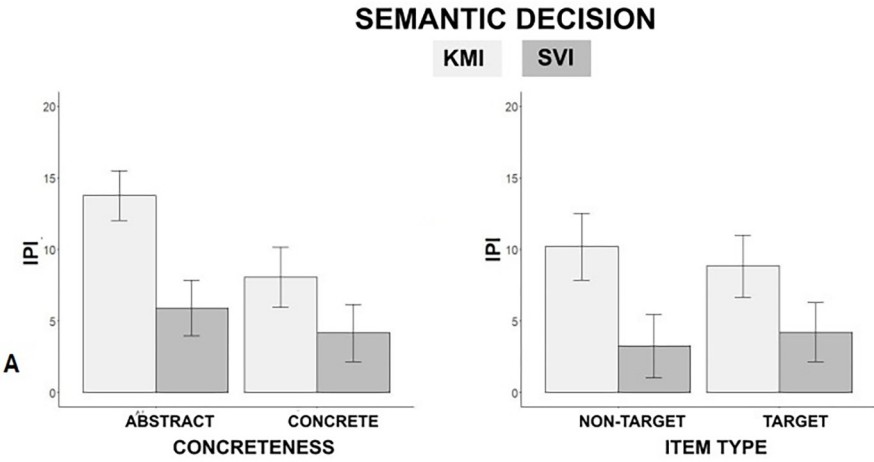

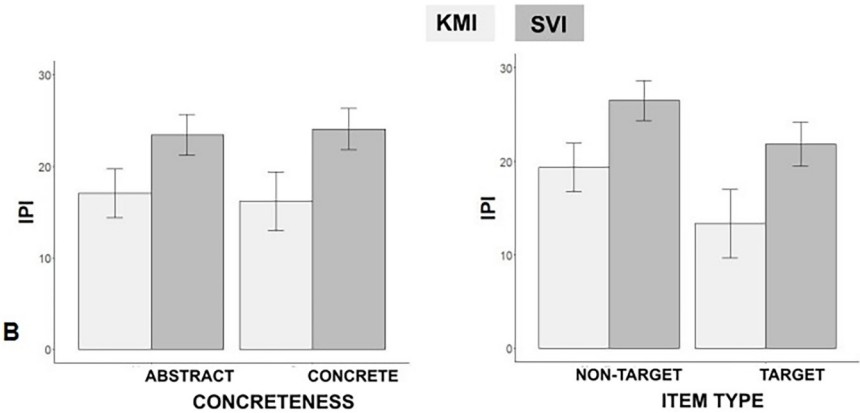

**Fig 2. Results of Experiment 1.** Reaction time (in ms) for both semantic categorization (A) and sentence-picture matching (B) tasks. Error bars indicate standard error of the mean.

"*Adrien creuse un trou dans son jardin*", "*Adrien digs a hole in his garden*") and 14 sentences included an abstract verb (e.g., "*Emma contemple son reflet dans le miroir*", "*Emma contemples her reflection in the mirror*"). Sentences shared a similar syntactic structure composed of a subject, verb, verb object, and locative adjunct. The number of syllables per sentence was controlled so that it did not differ significantly between the three categories of sentences. The task included a total of 84 trials divided into two blocks of 42 trials, thus each sentence was presented twice. All sentences were embedded in white noise with a signal-to-noise ratio of -8dB. The noise was included as previous studies showed that the motor system may be involved in speech perception, especially when task conditions are challenging (46,47). All stimuli were presented using E-Prime software (v2.0.10.353, E-Prime Psychology Software Tools Inc., Pittsburgh, USA). Manual responses were performed with the dominant right-hand and were made on a compatible E-prime SRBox. Accuracy and reaction time were recorded. Auditory stimuli were presented in a headphone with a comfortable sound level set by participants.

## Data analysis

To control for factors that could potentially influence participants' performance (i.e., musical and/or physical practice), independent-sample Mann-Whitney U tests were conducted. The

training group (i.e., KVI or SVI) was used as the independent variable, and weekly frequency of musical and/or physical practice served as dependent variables.

To evaluate participants' kinesthetic and visual imagery abilities, a two-factor mixed ANOVA was conducted. The training group (KMI or SVI) and subscale type (the visual or kinesthetic scale of the KVIQ-10) were computed as independent variables, and subscale scores were computed as dependent variables. As for MI abilities measured by the mental chronometry task, a correlation coefficient was computed for each group and the coefficients were compared using a Fisher's Z transformation.

To evaluate language comprehension performance, accuracy (% of correct responses) and response time (RT, in ms) were recorded for each trial in both language tasks. Only correct response trials were included in RT analyses. For the semantic categorization task, trials with RT of more or less than 2.5 standard deviations of the mean were scored as incorrect responses. For the sentence-picture matching task, the same criterion was used for the upper limit. However, the minimal cut-off was set at 100ms as it is known for being the time threshold for visual processing of images [65]. Missing data were excluded from the analysis. Three criteria were used to detect outliers: leverages, studentized deleted residuals (SDR), and Cook's distance. Observations with an SDR greater than 4 and/or a Cook's D value greater than 1 were considered extreme and were thus excluded [66]. Concerning RT analysis, models with maximal random effects structure were adopted [67, 68]. Two mixed-effect models were designed for each language comprehension task: one with the training group (i.e., KMI, SVI) and concreteness (i.e., abstract, concrete) as fixed factors and participants' and stimuli's intercepts as random factors, and the other with the training group (i.e. KMI, SVI) and item type (i.e., target, non-target) as fixed factors, and participants' and stimuli's intercepts as random factors. For accuracy analyses, two logistic mixed-effect models were computed for each language task with the same fixed and random factors as those for RT models. A Bonferroni correction was computed for all language task analyses to account for multiple comparisons (i.e., corrected alpha value set at 0.025). Data analyses were performed using the R environment version 4.0.3 (R Core Team, 2018), together with the lme4 [67], and the lmerTest [69] packages.

## Results

**Results from Step 1.** Results pertaining to musical and physical expertise suggested that participants in each training group did not significantly differ in terms of frequency of physical ($U$ = 187.5, $p$ = .401, with M = 2.08 hr, SD = 2.49 hr of practice per week for the KMI participants and M = 1.42 hr, SD = 1.73 hr of practice per week for the SVI group), or musical practice ($U$ = 169, $p$ = .732, with M = 0.813 hr, SD = 0.72 hr of practice per week for the KMI group, and M = 1.06 hr, SD = 3.44 hr of practice per week for the SVI group).

As for pre-training imagery abilities, results indicated that participants in each training group did not significantly differ in terms of MI abilities as measured by the mental chronometry task. Indeed, no significant main effect of the group was observed ($^z$ = -0.418, $p$ = .585 with $r(16)$ = 0.64, $p$ = .004 for the KMI participants and $r(16)$ = 0.54, $p$ = .002 for the SVI group). Regarding the KVIQ-10 scores, the same pattern of results was found with no significant differences between the two groups; $F(1,34)$ = 0.653, $p$ = .424 (with M = 13.33, SD = 4.52 for the KMI group and M = 12,88, SD = 3.95 for the SVI group) for both subscales: visual subscale (M = 16.5, SD = 4.94 for the KMI group; M = 14.94, SD = 5.16 for the SVI group) and kinesthetic scale (M = 13.33, SD = 4.52 for the KMI group and M = 12.89, SD = 3.95 for the SVI group).

No significant interaction effect was observed between the training group and subscale ($F(1,34)$ = 0.346, $p$ = .56). A significant main effect of the subscale type ($F(1,34)$ = 7.66, $p$ = .009)

was found, with higher average scores on the visual imagery scale (M = 15.72, SD = 5.04) compared to the kinesthetic scale (M = 13.11, SD = 4.19). Overall, results suggested no significant differences between the two training groups (i.e., KMI and SVI) on pre-training imagery measures.

**Results from Step 3: Performance on language comprehension tasks.** Participants successfully completed the semantic categorization task as suggested by accuracy scores (M = 94%, SD = 3.3). The logistic model including training group and concreteness factors revealed a significant main effect of group ($z$ = -2.27, $p$ = .023) with a better performance in the KMI group (M = 95%, SD = 2.3) than in the SVI group (M = 94%, SD = 2.5). A significant interaction between the training group and concreteness was observed ($z$ = 2.6, $p$ < .001) with a larger difference between KMI and SVI training groups for abstract words (M = 5%, SD = 7%) than for concrete words (M = -1%, SD = 2%). No significant effect was observed for the logistic model including training group and item type in terms of accuracy.

Analyses performed on RTs with the training group and concreteness as fixed factors were done on 35 participants as one participant had a Cook's D value over the cutoff point. A total of 284 RTs out of 4032 RTs were excluded for the semantic categorization task due to incorrect responses or outliers. Results showed a significant main effect of the training group ($b$ = .12, $t$(46.1) = 3.73, $p$ < .001). Post hoc analyses revealed a very large effect size ($d$ = 1.11) with an observed power given the corrected alpha being used of 91%. Participants in the KMI group were faster (M = 1096 ms, SD = 278 ms) than those in the SVI group (M = 1196 ms, SD = 308 ms). A significant main effect of concreteness was also observed ($b$ = -.14, $t$(52.5) = -5.81, $p$ < .001) with faster RTs for concrete words (M = 1086ms, SD = 266ms) compared to abstract words (M = 1271 ms, SD = 320 ms). The effect size pertaining to this main effect is considered extremely large ($d$ = 1.61), with an important statistical power of 95%. A significant interaction between training group and concreteness was also found ($b$ = -.04, $t$(2766.3) = -2.65, $p$ = .008). Post-hoc analyses revealed a small effect size regarding this interaction ($d$ = .01), with an only moderate achieved power (1−$\beta$ = .62) This interaction effect indicated that the difference in reaction times between the SVI and the KMI group was greater for abstract words (M = 140 ms, SD = 311 ms) than for concrete words (M = 84 ms, SD = 262 ms), see Fig 2A.

The model comparing training groups and item types (i.e., target vs. non-target) showed a significant effect of the training group ($b$ = .07, $t$(42.5) = 2.49, $p$ = .017; $d$ = .77; 1−$\beta$ = .23) with faster average RTs for participants in the KMI training group (M = 1044 ms, SD = 246 ms) compared to participants in the SVI training group (M = 1128ms, SD = 279 ms). No significant effect of item type and no interaction effects were observed ($b$ = 6.92, $t$(35.2) = 2.48, $p$ = .42, and $b$ = .02, $t$(1808) = 1.32, $p$ = .18, respectively).

For the sentence-picture matching task, data analyses were performed on a total of 35 participants, as data from one participant were excluded (mean RT exceeded group RT + 2.5 SD). A total of 664 out of 3024 RTs were excluded from data analyses due to incorrect responses or deviant observation. Participants successfully completed the sentence-picture matching task as suggested by accuracy scores (M = 78%, SD = 5%). No significant effect was observed in any of the logistic models carried on accuracy.

In terms of RTs, the model including training group and concreteness factors revealed no significant effect of training group ($b$ = .09, $t$(50.55) = 1.87, $p$ = .07) no significant effect of concreteness ($b$ = -.07, $t$(46.20) = -1.05, $p$ = .3) and no significant interaction ($b$ = .04, $t$(2209.2) = 1.3, $p$ = .19). The model including training group and item type showed a main effect of the training group ($b$ = .13, $t$(47.15) = 2.78, $p$ = .007) with faster RTs for participants in the KMI training group compared to participants in the SVI group (M = 1456 ms, SD = 659 ms for the KMI group and M = 1675 ms, SD = 716 ms for the SVI group). Post hoc analyses revealed a medium effect size ($d$ = .82) allowing us to achieve only a moderate statistical power given the

sample size and the corrected alpha (1−β = .30). Analyses did not show any significant main effect of item type (b = .13, t(31.31) = 1.72, p = .09), although average RTs were shorter for target items (M = 1450 ms, SD = 622 ms) compared to non-target items (M = 1686 ms, SD = 750 ms). There was no significant interaction between the training group and item type (b = .003, t(1449) = 0.082, p = .9, see Fig 2B).

## Discussion

A one-session training protocol was used to evaluate the effect of KMI training on language comprehension. More specifically, this experiment aimed at evaluating whether the KMI and SVI training protocols might induce differential effects on concrete and abstract verbs processing. In addition, transfer effects of the KMI training protocol were investigated throughout the type of items, with a greater training effect on target action verbs than on non-target previously anticipated. Overall, results on the two language tasks show that a single KMI training session can facilitate lexical-semantic access, as evidenced by significantly shorter RTs for the KMI group compared to the SVI training control group. This facilitation effect could be explained by a motor pre-activation induced by KMI but not by SVI. Regarding the verb concreteness, a stronger training effect for action verbs compared to abstract verbs was hypothesized. However, results indicated that the KMI participants were faster on all types of items, with a specifically greater training effect on abstract verbs. In line with this, the interaction between training group and concreteness indicated that the difference in performance between KMI and SVI participants was greater for abstract items in the semantic categorization task. Importantly, this effect could not be explained by individual imagery abilities, as results showed that both groups had comparable imagery abilities before training. Indeed, analysis on the KVIQ-10 and mental chronometry task showed no significant differences between groups before the training session. Moreover, the main effect of concreteness—resulting in shorter RTs for concrete items than for abstract items—supports previous results and can be explained by a difference in imageability between the two types of items (i.e., concrete words being highly-imageable, while abstract words are low-imageable [19, 70, 71]. Finally, results on the semantic categorization task did not reveal any processing differences across item types, which suggests that the potentially beneficial effect of KMI training on language comprehension is not limited to the specific trained items but is able to extend to other verbs. Results were less conclusive for the sentence-picture matching task, yet also partially supported a greater training effect for KMI participants than SVI participants. Different explanatory hypotheses can be put forward in light of the results. One way to explain the interaction effect that resulted in a greater training effect on abstract verbs for the KMI group revolves around a mostly beneficial effect of KMI training for words that are more difficult to process and less imageable. However, this explanation might be unlikely as results from the sentence-picture matching task only partially support our hypothesis, yet this task was maximizing degraded speech perception, which should have increased the difficulty associated with language processing. Conversely, it could be that the KMI training, and the consequent motor-cortex activation, resulted rather in an interference effect than a facilitation during the processing of action verbs, especially considering the effects of context on affordance activation. For instance, tasks and their different stimuli lead to different activations depending on the context and on the goal of the participant [72]. Affordances also vary depending on ownership [72], and the stimuli used in the sentence-picture matching task all included sentences referring to a third person represented in the pictures, which rather rely on the comprehension of action and action goals of others [72]. This first experiment supports our hypothesis that KMI training can improve language comprehension. However, we did not measure participants' performance on kinesthetic and visual imagery

following training. Therefore, the effects of both training protocols on imagery skills could not be measured. As the modulation of the motor cortex may depend on imagery quality [73], it is essential to be able to assess the effectiveness of the training protocols being proposed, by ensuring that the KMI group improves its MI skills more than the control group. In addition, as this experiment relied on a between-group design, we cannot exclude that the results might be related to interindividual differences in language performance rather than the beneficial effect of the KMI training on language comprehension. Hence, Experiment 2 aims at examining participants' improvement on language comprehension tasks and on visual and kinesthetic imagery measures with a pre/post-training procedure. For this purpose, an index of performance improvement was computed for each participant on language tasks. To further explore our hypotheses, a longer training was also implemented in an attempt to further engage the motor system and examine if it would result in greater improvements for the KMI group.

## Experiment 2

This second experiment aimed at confirming the hypothesized beneficial effect of KMI training on language comprehension performance, as compared to a SVI training. A pre-/post-training procedure was employed, taking into consideration interindividual differences in imagery abilities and language comprehension. Additionally, we adopted a longer training program, consisting of four training sessions spread over four consecutive days instead of an acute training session. This longer training aimed at enhancing the hypothesized recruitment of the motor system. The chosen procedure for Experiment 2 allowed us to analyze improvements from pre- to post-training on imagery measures, in order to examine the effects of our two training protocols on motor imagery skills. For each language task, we computed an index of performance improvement (IPI) so that we could quantify the specific improvement of participants in each group. Regarding the effectiveness of both training protocols and their applicability to our research questions, we hypothesized that participants from the KMI group will improve on both kinesthetic and visual motor imagery following their training program, which should result in a performance improvement on both KVIQ-10 subscales, with a probable greater effect on the kinesthetic component being expected given the instructions that were given to the participants during the KMI training. Moreover, we hypothesized that participants from the SVI group should keep relatively stable performance on the KVIQ-10, with an eventual improvement on the visual scale, considering the shared processes between visual motor imagery and static visual imagery. Similarly, a greater improvement on the mental chronometry task was hypothesized for the KMI group compared to the SVI one. Regarding performance on language tasks, we hypothesized that participants trained on KMI would show further improvements on both language tasks, compared to the SVI participants, and that these improvements would be greater for action verbs than abstract verbs and for actions previously imagined in the KMI training.

### Method

**Participants.**   Fifty-three undergraduate students (46 women and 7 men, mean age M = 20.2 years, SD = 2.9 years) from the University Grenoble Alpes participated in this second experiment. Participants were right-handed, native-French speakers with normal or corrected-to-normal vision. None of them reported having a history of neurological, psychiatric, or auditory pathologies. After Step 1 of the protocol, 26 students were assigned to the KMI group (mean age M = 19.8 years; SD = 1.7), and 27 were assigned to the SVI group (M = 20.7 years; SD = 3.7). After receiving an explanation of the procedures, participants gave written informed consent and were informed about the real objectives of the project at the end of the

experiment. Participants received extra credits for participating and the study was approved by the local ethical committee (CER Grenoble Alpes Avis-2018-12-11-2).

**Tasks and procedure.** The tasks and procedure were similar to those employed in the first experiment, except that the participants' imagery abilities and performance on language comprehension tasks were assessed before and after training. Moreover, participants received a longer training, consisting of four training sessions spread over four days instead of a single training session.

*Step 1*: *Pre-training assessment*. Participants filled in a survey containing questions about sport and music practice (i.e., some factors that could affect MI abilities [58], and completed the Kinesthetic and Visual Imagery Questionnaire (KVIQ-10; [59]. Then, they completed the mental chronometry task as well as the semantic categorization and the sentence-picture matching task used in Experiment 1. This experimental session lasted about an hour.

*Step 2*: *Training*. After completing the first step, participants were randomly assigned to the KMI training group or the SVI training group. The training protocols were identical to those in the first experiment. Participants took part in four training sessions on the four consecutive days following Step 1. Each training session lasted about 15 minutes.

*Step 3*: *Post-training assessment*. The third step of the experiment took place after the last training session. Participants were asked to re-complete the KVIQ-10, the mental chronometry task, and the two language comprehension tasks. This last experimental session lasted about one hour.

## Data analysis

**Pre-training imagery abilities.** To control for factors that could potentially influence participants' performance, namely, musical and/or physical practice, two independent-samples Mann-Whitney U tests were conducted. The training group was used as an independent variable, and weekly frequency of musical and/or physical practice served as dependent variables.

To evaluate participants' kinesthetic and visual imagery skills, a three-factor mixed ANOVA was conducted, with training group (KMI or SVI), subscale type (the visual or kinesthetic scale of the KVIQ-10), and time of assessment (pre-, or post-training) as independent variables. Participants' scores on each subscale were used as dependent variables. As for MI abilities measured using the mental chronometry task, the temporal correlation between imagined and executed movements was computed for each group. A between-group comparison was performed using a Fisher's Z transformation. In addition, results were analyzed using the Dunn and Clark test to evaluate the evolution of the temporal correlation between imagined and executed movements throughout time for participants in the same group.

**Performance on language comprehension tasks.** Accuracy (% of correct responses) and response times (RT) were recorded for each trial in both language tasks. Only correct response trials were included in RT analyses. Similar to the criteria in Experiment 1, for the semantic categorization task, trials with RT of more or less than 2.5 standard deviations of the mean were scored as incorrect responses. For the sentence-picture matching task, the same criterion was used for the upper limit. As in Experiment 1, the minimal cutoff was set at 100ms as it is known for being the time threshold for visual processing of images [65]. Missing data were excluded from the analysis. Three criteria were used to detect outliers: leverages, studentized deleted residuals (SDR), and Cook's distance. Observations with an SDR greater than 4 and/or a Cook's D value greater than 1 were considered extreme and were thus excluded.

We computed an Index of Performance Improvement (IPI = ([pre-training RT–post-training RT] / pre-training RT) *100) for each participant and each type of item (i.e., target, non-target, abstract) to measure performance improvement from pre- to post-training. A positive

index indicated a performance improvement, with RT decreasing in the session following training compared to the pre-training session. Conversely, a negative index indicated a decrease in performance. Two mixed effect models were designed for each language comprehension task using IPI as the dependent variable with (1) the training group (i.e., KMI, SVI) and concreteness (i.e., abstract, concrete) as fixed factors; participants' and stimuli's intercept as random effect factors and (2) the training group (i.e., KMI, SVI) and item type (i.e., target, non-target) as fixed factors and participants' and stimuli's intercept as random effect factors. Due to a ceiling effect, accuracy scores were not analyzed. We applied a Bonferroni correction for all language task analyses and set the alpha value at 0.025.

## Results

**Pre-training imagery abilities.** No significant differences were observed between the two training groups in terms of frequency of physical ($U$ = 379, $p$ = .613, with M = 3.25 hr, SD = 2.84 hr of practice per week for the KMI group and M = 3.15 hr, SD = 3.19 hr of practice per week for the SVI group), or musical practice ($U$ = 396, $p$ = .228, with M = 0.92 hr, SD = 2.32 hr of practice per week for the KMI group, and M = 0.08 hr, SD = 0.38 hr of practice per week for the SVI group).

For the mental chronometry task, no significant difference was observed ($\wedge z$ = -0.22, $p$ = .82) between the two group in terms of MI abilities in the pre-training phase (with $r$(26) = 0.53, $p$ = .005 for the KMI participants and $r$(27) = 0.578, $p$ = .002 for the SVI group). The Dunn and Clark tests revealed there were no significant differences pertaining to the temporal correlation between pre-and post-training evaluations within each group ($\wedge z$ = -0.86, $p$ = .39 for the KMI participants and $\wedge z$ = -0.9094, $p$ = .36 for the SVI participants).

Analysis conducted on the subjective imagery measure in Steps 1 and 3 showed that there was not a significant main effect of the group on KVIQ-10 scores ($F$(1,51) = 0.084, $p$ = .774). There was no significant interaction between training group and assessment time ($F$(1,51) = 1.101, $p$ = .299). However, the interaction between subscale type and assessment time was significant ($F$(1,51) = 8.722, $p$ = .005; $d$ = .81). Furthermore, a significant interaction between training group, subscale type, and assessment time, ($F$(1,51) = 19.03, $p$ < .001; $d$ = 1.20) was also observed. This interaction indicated that the KMI group significantly improved on the kinesthetic subscale following training (average score pre-training M = 12.23, SD = 5.17, and average score post-training M = 16.12, SD = 4.83), but not on the visual subscale (average score pre-training M = 18.81, SD = 4.27, and average score post-training M = 18.27, SD = 4.52). Participants in the SVI group significantly improved on both subscales the kinesthetic (average score pre-training M = 13.41, SD = 4.42 and average score post-trading M = 15.48, SD = 4.40) and the visual (average score pre-training M = 17.33, SD = 4.52 and average score post-training M = 20.26, SD = 3.82).

Overall, these results suggested that participants had similar imagery abilities before training and did not improve on the mental chronometry task. While KMI participants improved in terms of ratings on the kinesthetic scale of the KVIQ-10 only, participants in the SVI group improved on both the kinesthetic and visual scales.

**Performance on language comprehension tasks.** In the semantic categorization task, participants successfully completed the task as suggested by accuracy scores (M = 88,5%, SD = 10.4% in pre-training; M = 91.5% and SD = 9.2% in post-training). No analyses were conducted on accuracy due to a ceiling effect.

Analyses in terms of RTs using IPI, using data from Step 1 (pre-training) and Step 3 (post-training) were performed on 51 participants after the exclusion of two outliers following our a priori Cook's D criteria. Specifically, 619 out of 6048 RTs from the pre-training and 467 out of

5936 RTs from the post-training semantic categorization data were excluded due to incorrect responses or outliers. The model including the training group and concreteness as fixed factors and IPIs of RTs as dependent variable revealed a significant main effect of training group ($b$ = 6.47, $t$(87.8) = -2.8, $p$ = .006; $d$ = .60) suggesting a greater improvement for the KMI group (M = 9.8%, SD = 29.3%) than for the SVI group (M = 4.7%, SD = 28.9%). The model also revealed a significant main effect of concreteness ($b$ = -1.12, $t$(98.2) = -3.19, $p$ = .001; $d$ = .65) with participants having a greater average IPI for abstract words (M = 9.72%, SD = 27.08%) than for concrete words (M = 6.06%, SD = 30.02%). We also observed a marginally significant interaction between the training group and concreteness ($b$ = 1.47, $t$(3949) = 1.98, $p$ = .05; $d$ = .06). This marginal effect suggests that the difference between KMI and SVI training groups in terms of IPI was greater for abstract (M = 7.8) than for concrete verbs (M = 3.9%, see Fig 3).

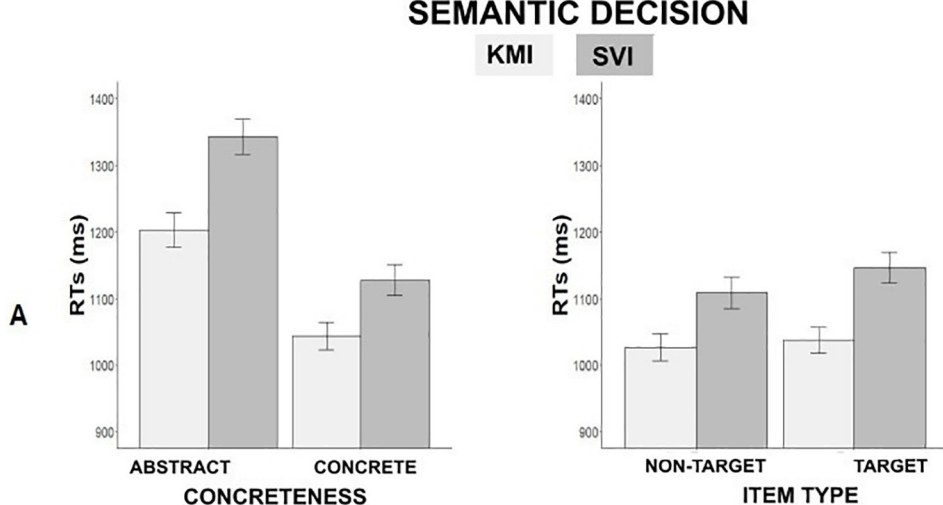

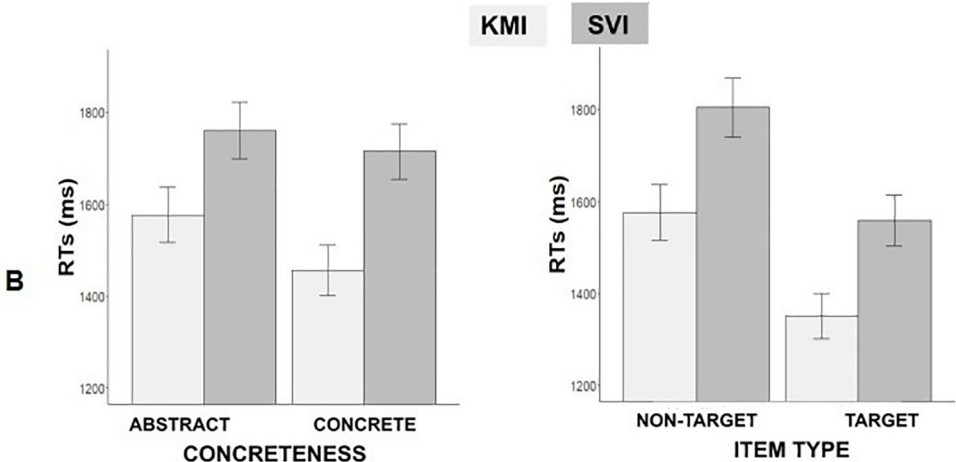

**Fig 3. Results of Experiment 2.** Index of Performance Improvement (IPI in %, computed as: IPI = ([pre-training RT–post-training RT] / pre-training RT) ∗100)) for both semantic categorization (A) and sentence-picture matching (B) tasks. Error bars represent standard errors of the mean.

Post hoc power analyses given corrected alpha value, sample size, and effect size revealed an achieved power of 94%, 15%, and 5% for the three effects presented above, respectively.

The model including the training group and the item type (i.e., target vs. non-target) as fixed factors and IPIs as dependent variable showed a significant effect of the training group ($b$ = -6.81, $t(76.6)$ = -2.35, $p$ = .01; $d$ = .52), with participants in the KMI training group demonstrating a greater average IPI (M = 9.5%, SD = 28.5%) compared to participants in the SVI-training group (M = 4.8%, SD = 27.3%). Analyses revealed an absence of item type main effect ($b$ = -1.55, $t(53.6)$ = -0.82, $p$ = .41) as well an absence of interaction between training group and item type ($b$ = 2.41, t(2593.7) = 0.79, $p$ = .43).

For the sentence-picture matching task, participants successfully performed the task as suggested by accuracy scores (M = 76.2% and SD = 10.6% at pre-training; M = 83.6% and SD = 8.9% at post-training). One outlier, according to Cook's D value, was detected and excluded, and IPI analyses were performed on data from 52 participants. Specifically, 1011 out of 4452 RTs were excluded due to incorrect responses or outliers.

For analyses in terms of RTs, the mixed-effects model including training group and concreteness showed no significant main effect of the training group ($b$ = 6.47, t(74.8) = 1.2, $p$ = .02) although participants in the SVI training group showed a greater IPI (M = 23.85%, SD = 32.48%) than those in the KMI training group (M = 16.48%, SD = 43.79%). Moreover, no significant effect of concreteness and no significant interaction were observed ($b$ = -1.13, $t$ (62) = 1.92, $p$ = .05; $b$ = 1.47, $t(2162.4)$ = 0.55, $p$ = .58, respectively). The analysis for the model with training group and item type as fixed factors was conducted on 52 participants, as one outlier was identified and removed. This model revealed a significant effect of training group ($b$ = 8.76, t(86.2) = 2.63, $p$ = .01; $d$ = .57) participants in the SVI training group (M = 24.1%, SD = 32.7%) showing greater average IPI compared to participants in the KMI training group (M = 16.2%, SD = 46.5%). The main effect of item type was not significant ($b$ = 6.16, $t(54.42)$ = 2.03, $p$ = .05; $d$ = .57) with a greater IPI for non-target words (M = 23%, SD = 34.6%) than for target words (M = 17.5%, SD = 44.8%). No significant effect of the interaction between training group and item type was observed ($b$ = -1.79, $t(2124.2)$ = -0.54, $p$ = .59) (see Fig 3).

Overall, these results suggest that the performance of participants in the KMI training group improved more from one evaluation time to another in the semantic categorization task, as compared to the performance of participants in the SVI training group (see Fig 3A). However, this was not specific to item type. In the sentence-picture matching task, participants trained on SVI demonstrated a greater improvement than participants trained on KMI (see Fig 3B).

## Discussion

The second experiment employed a pre/post-training procedure in efforts to address some of the limitations of Experiment 1. The main objective was to assess and compare to what extent participants in two different training groups would improve on language tasks and imagery measures following their respective training protocols. Results provided evidence that the KMI training participants demonstrated a greater performance improvement on the semantic categorization task compared to the SVI training group. Only a marginally significant interaction effect between the training group and concreteness was revealed, indicating that KMI can improve conceptual processing of verbs overall, with a tendency towards improving the processing of abstract items specifically. A different pattern of results was observed for the sentence-picture matching task whereby the SVI group seems to have been favored over the KMI group. This pattern might be due to task demands (i.e., visual search), since the sentence-picture matching task required visuospatial and visual search and processing skills. This task was

included as a more ecological measure of language comprehension compared to the semantic categorization task. Indeed, it requires the processing of more complex linguistic information, as sentences were used instead of isolated verbs. The task also relied on a different decision-making process, whereby participants had to find the relationship between the sentences' meaning and the presented visual scenes. Results show that the SVI group improved more than the KMI group on this task, regardless of item type. Several factors could explain this result. Firstly, the sentence-picture matching task used here required a choice between two different complex visual scenes. As a result, the success of this task depends on visual search abilities, which might have been trained as part of the SVI training protocol. With respect to the effectiveness of both training programs, the pre/post-training procedure carried out using the KVIQ-10 indicated that both KMI and SVI participants had improved kinesthetic abilities and that the SVI group further developed their visual imagery abilities, which was an unexpected outcome. Moreover, both static visual imagery (in this case, of body parts and movements), and kinesthetic imagery are components of MI [58], which could explain why the SVI group managed to substantially make progress on both KVIQ-10 subscales, and consequently, demonstrated greater improvements on the sentence-picture matching task. This calls into question the use of SVI training as a proper control training. In addition, it raises the question as to whether the sentence-picture matching task was the appropriate measure of language comprehension to be used in the present study, and in particular because of its inherent visual components. As discussed in Experiment 1, it is important to consider that the sentence-picture matching task featured a third-person individual executing an action or illustrating an abstract concept, concepts which are usually low-imageable [19, 70]. The illustration of a third person executing an action may have activated affordances different from those activated by the KMI training, in which participants are asked to perform first-person imagery. This may have resulted in an interference effect for the KMI participants [72], while the SVI training may have indirectly facilitated the imageability of abstract concepts presented in the sentence-picture matching task through visual imagery skills improvement. These abilities might have been helpful for the performance on the sentence-picture matching task. Moreover, the results in this second experiment showed that more improvement was observed for abstract items compared to action items. The latter finding possibly indicated that our action items required relatively less effort compared to abstract items, as previously shown in Experiment 1.

## General discussion

The objective of the present study was to assess the impact of motor-imagery training on language comprehension. We hypothesized that participants in the Kinesthetic Motor Imagery (KMI) training would exhibit, on average, better performance on language comprehension tasks than participants in the Static Visual Imagery (SVI) training control group. Several studies have examined the effect of sensorimotor expertise on semantic processing of action-specific semantic concepts [35, 37, 74]. To the best of our knowledge however, there are currently no studies that directly examined the effect of MI training alone (i.e., not combined with execution or observation of action) on language comprehension tasks. In the present study, the effects of a one-session (Experiment 1) and a four-session (Experiment 2) training protocol on lexico-semantic processing was explored.

Overall, the results seem in favor of the hypothesis that a KMI training can facilitate lexico-semantic access when compared to a control training based on SVI. Based on the pattern of results and what is known about the motor system and Motor Imagery (MI), it could be hypothesized that the KMI training led to a greater activation of sensorimotor systems than the SVI training did.

The literature showed that MI activates the motor system in a somatotopic manner similar to action execution [51, 75] and that MI can induce neuroplasticity [50, 76]. An explanatory assumption could thus be that the present KMI training resulted in a spread of activation throughout the motor system, which would act in a priming-like manner if language is indeed grounded in sensorimotor systems, as proposed by the embodied cognition framework. Considering that no differences were observed between target and non-target items, it is likely that KMI-induced neuroplasticity effects [36, 50, 76] spread to larger motor system areas, rather than effector-specific areas. In addition, as the improvement of behavioral performance was observed not only for concrete action verbs but also for abstract verbs, it is possible that the training-neuroplasticity effects induced a global increase in activity within the motor cortex, namely in higher-order motor areas that are not organized somatotopically, resulting in behavioral modulations that were not category-specific.

It is important to consider that KMI was previously found to increase the activation of the sensorimotor system [75, 77, 78]. Consequently, one may postulate that the KMI training resulted in faster RTs through a pre-activation of the primary motor cortex hand area and did not affect lexico-semantic access per se. However, if the difference between SVI and KMI training is only explained by the motoric component of the response, we believe that we would not have observed the interaction between the type of training and the type of verb to be processed. Indeed, we observed that the difference of RTs between KMI and SVI was greater for abstract items. If the effect was solely due to faster manual responses, we would expect only a main effect of the training condition. Some previous studies found that motor-based training protocols, such as action observation or execution, can improve abstract verb processing (22). It is therefore conceivable that the effects of the KMI training were also visible for the abstract items. Furthermore, as observed in previous studies [19, 70], it seems that abstract items might be harder to process, potentially due to the high-imageability of concrete items as compared to low-imageable abstract items. It is interesting to note that some experiments show that the motor system is more involved in language processing when the processing is considered difficult [79–81]. Thus, the interaction that we observed may be explained by greater involvement of the motor system in the processing of abstract verbs.

Visual imagery training can induce an increase in attentional processes [36, 44, 82], allowing for an improvement in performance that is not due to motor system activation. Additionally, it can improve perceptual learning and performance on tasks requiring visuospatial skills [82], such as the sentence-picture matching task used in the present study. This can further explain why, in Experiment 2, the SVI group's performance on the sentence-picture matching task improved significantly compared to the KMI group. Moreover, while the videos used in the training were silent and the pictures were not labeled, it remains possible that participants in the KMI group did silently label the actions they saw during the training, consequently retrieving the semantic concepts during the training, resulting in faster reaction times compared to the SVI group. Nevertheless, since there was no significant effect of item type or an interaction effect between item type and training condition, these explanations seem unlikely. Furthermore, during the debriefing, none of the participants reported having identified any of the items in the tasks as being actions they had imagined in the KMI training phase. Additionally, there were no significant differences between the two groups in Experiment 2 on the sentence-picture matching task, meaning that the faster reaction times for the KMI group in Experiment 1 or for the semantic categorization task in Experiment 2 were unlikely due to an increased familiarity of the KMI group with the action-verb stimuli used in the task. Moreover, other motor-based training aiming to improve language processing showed more global language improvements that are not only limited to action words [52, 53].

Some of the results may in fact be driven by the demands of the sentence-picture matching task. It could be due to the affordances activated by the verbs in the sentences, by the actions depicted in the picture, or by the contextual elements in the pictures [72]. Nevertheless, while the sentence-picture matching task does involve semantic retrieval [19, 56, 83], it also may induce semantic interference in participants when pictures and words belong to the same semantic category [56]. The present KMI protocol may have activated semantic representations of body parts related to the actions verbs, which may have exacerbated this interference effect, as semantic-to-motor priming has been documented [84]. In our study, we used sentences rather than words, but the sentences were controlled on various psycholinguistic variables and participants' responses were related to the verb in the sentence. Indeed, embedding verbs in sentences may complicate a sentence-picture matching task compared to words only, as they require accessing additional semantic orders [56, 83]. The semantic categorization task requires access to semantic information about semantic membership but does not necessarily require access to fine-grained semantic level categories [56, 83]. By contrast, sentence-picture matching tasks necessitate the navigation of visual elements as well as making fine-grained semantic decisions about a set of visual elements that could be associated with the word, or sentence [19, 56, 83]. It may also be that a visual imagery training may have facilitated the subsequent visual search required in the sentence-picture semantic task. In Experiment 2, a longer training was required, which may explain why the main effect of the group (more improvement for the SVI group) in the sentence-picture matching task was found in Experiment 2, while the opposite pattern of results (i.e., slower RTs for the SVI group) was found in Experiment 1, after an acute training session.

It has been brought to our attention by the reviewers and the resources they provided that KMI training, followed by tasks that include some action verbs, may result in an interference rather than a facilitation of the processing of action words [72]. In Experiment 1, participants in the KMI group were overall faster than participants in the SVI group, and this difference was significantly greater for abstract verbs compared to concrete verbs in the semantic categorization task. The participants in the KMI group were faster in the processing of target verbs compared to non-target, untrained verbs, possibly indicating that the KMI training interfered with the processing of the untrained action verbs specifically. In Experiment 2, there were no interactions within the semantic categorization task, and the SVI group demonstrated better improvement in reaction times in the sentence-picture matching task. It is possible that an interference occurred between the areas activated in the motor cortex, and the action-information conveyed by sentences and pictures in the sentence-picture matching task.

The longer training protocol implemented in Experiment 2 could have induced neuroplasticity effects that affected performance on language comprehension in a different pattern from Experiment 1. Indeed, it has been shown that a single session of MI practice induces use-dependent plasticity, driven by an increase of corticospinal excitability in agonist muscles, and this excitability was shown to return to baseline levels when assessed 30 minutes later [50]. In Experiment 1, the affordances activated by the KMI training may have been incongruent with the affordances activated by some of the action verbs included in the language tasks, namely, the sentence-picture matching task that included more contextual elements in its material. This might have been illustrated by the interaction effect observed, which was opposite to our previously-hypothesized interaction of a specific facilitatory effect of the KMI training in action verbs compared to abstract verbs. For the sentence-picture matching task, it seems that the absence of pre-activation resulted in an advantage for the SVI participants, who had shorter RTs compared to the KMI participants. Since there was still a main effect of the group in one task but not the other task, it lessens the likelihood that the effect observed in Experiment 1 was due to faster manual responses induced by KMI hand-action training. Moreover,

abstract verbs seem to be grounded in sensorimotor areas, but to a lesser extent compared to action verbs [15] It could explain the interaction between group and verb type (i.e., target, non-target), as well as group and concreteness (i.e., abstract, concrete) of Experiment 1 in the semantic decision task, as abstract verbs did not induce interference, unlike untrained action verbs. Nevertheless, both facilitation and interference (or inhibition) effects highlight the bidirectional relationship between language and action.

Finally, while our study included native French speakers only, we cannot disregard the potential role that bilingualism may have played. We did not examine whether any of our participants were early simultaneous bilingual (i.e., acquired French in parallel with a second native language), and influenced by another culture. Culture also appears to play a role in the extent of grounding of language. Ghandhari et al. [85] showed in their behavioral study that native Italian participants exhibited a facilitation effect when processing a concrete sentence that was congruent with a video displaying a movement, while native Persian participants exhibited an inhibition effect in the same congruent and concrete condition. Moreover, Italian participants were overall faster for concrete sentences (i.e., concreteness effect) regardless of congruency, whereas Persian participants presented inhibition particularly for concrete sentences in congruent trials, but processing times did not differ based on concreteness in incongruent trials [85]. Likewise, in a TMS study comparing the semantic and motor resonance effects of first language versus second language, participants categorized verbs as abstract or concrete while TMS was applied to the hand motor cortex at varying time intervals following stimulus presentation. Results revealed that semantic resonance occurred at early stages of semantic processing (TMS applied at 125 ms post-stimulus) for first-language words but not second-language words, while motor resonance induced by TMS occurred at slightly later stages of semantic processing (275 ms post-stimulus) for second-language words. These results support the notion that action and language interact at early stages of word processing, and first-language and second-language are differently embodied [86]. Nevertheless, studies show that children who acquire two languages from birth learn language in a manner comparable to monolingual children, whereas highly proficient late bilinguals seem to have a more grounded representation of their second language ([87] for a review).

## Conclusion

The present study showed for the first time to our knowledge that a KMI-only training, known to increase sensorimotor activation, can facilitate lexico-semantic processing, alluding to the presence of functional links between sensorimotor and language areas. Previous studies have investigated the role of motor experience and action observation (e.g., [52]), or motor imagery combined with another motor function [53], while ours was the first, to our knowledge, to assess the role of a KMI-only training on language comprehension. A motor-imagery training seems to have carry-over effects on a range of action concepts that extend beyond the trained concepts only. Future protocols could employ causal methods such as TMS to further explore the nature of the link between the motor system in general—and MI in particular—and the language system. They could also employ neuroimaging techniques to evaluate the spatial and temporal characteristics of the neural underpinnings of the mechanisms in question. Results concerning the link between the sensorimotor and language system could have implications for future rehabilitation protocols for language in patients with various pathologies, such as stroke patients [45, 53, 88]. These results can provide insights to designing rehabilitation methods based on principles of training, compensatory mechanisms, brain plasticity evidence, and exploitation of residual abilities [53, 89, 90]. Indeed, MI presents several advantages that make it worthy of consideration for a complementary rehabilitation method of language deficits. For

example, by relying on their residual abilities, such as mental imagery practice, patients for whom covert sensorimotor abilities are affected, could improve their motor functioning while potentially activating the language system.

## Acknowledgments

We thank all of the participants who agreed to participate in the present study. We also thank the reviewers for their constructive suggestions and comments that improved the presentation of our study.

## Author Contributions

**Conceptualization:** Camille Bonnet, Richard Palluel-Germain, Marcela Perrone-Bertolotti.

**Data curation:** Camille Bonnet, Mariam Bayram, Richard Palluel-Germain, Marcela Perrone-Bertolotti.

**Formal analysis:** Camille Bonnet, Mariam Bayram, Samuel El Bouzaïdi Tiali, Richard Palluel-Germain, Marcela Perrone-Bertolotti.

**Funding acquisition:** Marcela Perrone-Bertolotti.

**Investigation:** Camille Bonnet, Richard Palluel-Germain, Marcela Perrone-Bertolotti.

**Methodology:** Camille Bonnet, Mariam Bayram, Samuel El Bouzaïdi Tiali, Richard Palluel-Germain, Marcela Perrone-Bertolotti.

**Project administration:** Marcela Perrone-Bertolotti.

**Resources:** Richard Palluel-Germain, Marcela Perrone-Bertolotti.

**Software:** Sylvain Harquel, Marcela Perrone-Bertolotti.

**Supervision:** Richard Palluel-Germain, Marcela Perrone-Bertolotti.

**Validation:** Mariam Bayram, Samuel El Bouzaïdi Tiali, Florent Lebon, Sylvain Harquel, Richard Palluel-Germain, Marcela Perrone-Bertolotti.

**Visualization:** Mariam Bayram, Samuel El Bouzaïdi Tiali, Richard Palluel-Germain, Marcela Perrone-Bertolotti.

**Writing – original draft:** Camille Bonnet, Mariam Bayram, Samuel El Bouzaïdi Tiali, Florent Lebon, Sylvain Harquel, Richard Palluel-Germain, Marcela Perrone-Bertolotti.

**Writing – review & editing:** Camille Bonnet, Mariam Bayram, Samuel El Bouzaïdi Tiali, Florent Lebon, Sylvain Harquel, Richard Palluel-Germain, Marcela Perrone-Bertolotti.

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
