## [Decision Letter · Decision Letter 0]

29 Mar 2022

PONE-D-22-03552Kinesthetic Motor-Imagery Training Improves Performance on Lexical-Semantic AccessPLOS ONE

Dear Dr. Perrone-Bertolotti,

Thank you for submitting your manuscript to PLOS ONE. After careful consideration, we feel that it has merit but does not fully meet PLOS ONE’s publication criteria as it currently stands. Therefore, we invite you to submit a revised version of the manuscript that addresses the points raised during the review process.

We look forward to receiving your revised manuscript.

Kind regards,

Victor Frak, MD, Ph.D

Academic Editor

PLOS ONE

Journal Requirements:

2. We note that Figure 1 includes an image of a participant in the study. 

Additional Editor Comments:

Dear Marcela,

You will find enclosed the comments of two referees about your work.

They both found it very interesting.

I encourage you to take these recommendations into account and send us the article again with the proposed modifications.

Many thanks for considering PLOS One to publish your research.

Best regards,

Victor Frak

Reviewers' comments:

Reviewer's Responses to Questions

**Comments to the Author**

1. Is the manuscript technically sound, and do the data support the conclusions?

Reviewer #1: Yes

Reviewer #2: Yes

2. Has the statistical analysis been performed appropriately and rigorously? 

Reviewer #1: Yes

Reviewer #2: N/A

3. Have the authors made all data underlying the findings in their manuscript fully available?

Reviewer #1: Yes

Reviewer #2: Yes

4. Is the manuscript presented in an intelligible fashion and written in standard English?

Reviewer #1: Yes

Reviewer #2: Yes

5. Review Comments to the Author

Reviewer #1: I find the paper very interesting and well written. In two different experiments, it has been shown the impact of motor imagery on lexical-semantic access, specifically it has been demonstrated that a kinesthetic motor imagery training improves language comprehension.

Please find below my comments:

Introduction

The introduction lack of some references about behavioural and neurophysiological studies on the cross-talk between action and language, which sometimes are in contrast. Indeed some studies have highlighted a facilitation, some others an interference effect in case of congruency between the meaning of the sentence and the performed action.

I suggest to look at:

Scorolli, C., & Borghi, A. M. (2007). Sentence comprehension and action: Effector specific modulation of the motor system. Brain research, 1130, 119-124.

Gianelli, C., & Dalla Volta, R. (2015). Does listening to action-related sentences modulate the activity of the motor system? Replication of a combined TMS and behavioral study. Frontiers in psychology, 5, 1511.

Sato, M., Mengarelli, M., Riggio, L., Gallese, V., & Buccino, G. (2008). Task related modulation of the motor system during language processing. Brain and Language, 105(2), 83–90.

Miller, J., Brookie, K., Wales, S., Wallace, S., & Kaup, B. (2018). Embodied cognition: Is activation of the motor cortex essential for understanding action verbs?. Journal of Experimental Psychology: Learning, Memory, and Cognition, 44(3), 335.

Chersi, F., Thill, S., Ziemke, T., & Borghi, A. M. (2010). Sentence processing: linking language to motor chains. Frontiers in neurorobotics, 4, 4.

The authors does not mention any reference concerning the relation between action and abstract/concrete language, although such dichotomy is matter of investigation of the studies and formulate some predictions.

Please look at:

Scorolli, C., Binkofski, F., Buccino, G., Nicoletti, R., Riggio, L., & Borghi, A. M. (2011). Abstract and concrete sentences, embodiment, and languages. Frontiers in Psychology, 2, 227.

Ghandhari, M., Fini, C., Da Rold, F., & Borghi, A. M. (2020). Different kinds of embodied language: A comparison between Italian and Persian languages. Brain and Cognition, 142, 105581.

Desai, R. H., Conant, L. L., Binder, J. R., Park, H., & Seidenberg, M. S. (2013). A piece of the action: modulation of sensory-motor regions by action idioms and metaphors. NeuroImage, 83, 862-869.

Sakreida, K., Scorolli, C., Menz, M. M., Heim, S., Borghi, A. M., & Binkofski, F. (2013). Are abstract action words embodied? An fMRI investigation at the interface between language and motor cognition. Frontiers in human neuroscience, 125.

Scorolli, C., Binkofski, F., Buccino, G., Nicoletti, R., Riggio, L., & Borghi, A. M. (2011). Abstract and concrete sentences, embodiment, and languages. Frontiers in Psychology, 2, 227.

Sample size:

The authors does not justify with any reference to the literature or power analysis the sample size of the both the experiments. Please provide a justification.

Could you please report and translate the items in English of KIVQ-10?

Why did you opt for a between-subjects design in the first experiment, instead of a within-subject design in which participant would have performed in different days?

Lines 214: “presented” repeated twice

Data Analysis

Could you please report the number of RTs discarded in both the experiments?

Could you explain the Cook D criteria?

Did you adopt as criteria the stepwise procedure in your models selection? Starting with the model with the more complex random structure? See the following reference:

Barr, D. J., Levy, R., Scheepers, C., & Tily, H. J. (2013). Random effects structure for confirmatory hypothesis testing: Keep it maximal. Journal of memory and language, 68(3), 255-278.

Results & Discussion

I see that in the first experiment you got that participants were faster with abstract as compared with concrete verbs. Why in your opinion this is the case? Could you please argument more about that in the discussion?

Did you consider the influence of the culture in explaining your results? Were all your participants native French? Was someone bilingual?

I recommend to consider these factors as it is suggested in Ghandhari, M., Fini, C., Da Rold, F., & Borghi, A. M. (2020). Different kinds of embodied language: A comparison between Italian and Persian languages. Brain and Cognition, 142, 105581.

It might depend on the level of integration between language and action, it might depend on the perceptual strength associated with the verbs.

One possibility is that abstract verbs “might occupy less the sensorimotor system “ and their categorization might be favoured, to better clarify , I mean that it might be possible that a congruency between action and concrete verbs is expressed by a interference effect instead of facilitation.

Finally, Here in you can find a broad perspective on the relation between the motor system and the language, especially the abstract one. Mazzuca, C., Fini, C., Michalland, A. H., Falcinelli, I., Da Rold, F., Tummolini, L., & Borghi, A. M. (2021). From affordances to abstract words: The flexibility of sensorimotor grounding. Brain Sciences, 11(10), 1304.

Reviewer #2: The manuscript entitled "Kinesthetic Motor-Imagery Training Improves Performance on Lexical-Semantic Access" investigates the effect of Kinesthetic Motor-Imagery training on semantic processing. I very much enjoyed reading this manuscript and I believe that the research presented here would fit well within the scope of this journal. Within an embodied framework, the topic is timely. However the theoretical motivations could be better justified and the analytical scaffolding would benefit from a more thorough consideration of the literature.

Major Remarks

Introduction

Overall, the theoretical motivation behind this study, as well as the theoretical implications, could be made more explicit. The introduction would benefit from an additional discussion on to the action-word processing (and possibly learning) literature. This would enrich the theoretical background and better justify the motivation for this work.

Given that the authors are interested in the effect of training involving the motor system on language processing, I would suggest they discuss studies using language learning paradigms to test the influence of encoding language with action or while stimulating or inhibiting motor areas. For instance, applying TMS in motor areas in conjunction with hand action-based word learning showed a causal involvement of the motor cortex in action word acquisition and even indicated rapid wide-spread microstructural changes in the language system linked to the motor cortex function (Vukovic & Shtyrov, 2019; Vukovic et al., 2021). In another TMS study, Mathias and colleagues (2021) examined training on foreign language words with sensorimotor (gestures) or visual (pictures) enrichment. They measured learning using a translation task while either repetitive TMS was applied bilaterally over the primary motor cortex, or a sham stimulation was applied. Perturbation from repetitive TMS lead to slower translations of sensorimotor-enriched words but not sensory-enriched words.

There are also several fMRI studies that look at encoding while observing or performing actions or gestures (Fargier et al., 2012, Macedonia et al., 2019; Macedonia & Mueller, 2016).

One of the authors’ hypotheses involves a contrast between abstract and action verbs (i.e., that the KMI group will perform better on action compared to abstract verbs). As such, the introduction would benefit from a section on the contrast between abstract and concrete language to justify this hypothesis. Some argue that abstract and concrete are represented differently (i.e., verbal-language vs. perceptual) (Montefinese, 2019; Wang et al., 2010; but see Ponari et al 2018). In addition, it has been shown that concrete words are processed more quickly than abstract words (for a review, see Huang & Federmeier, 2015), sometimes referred to as the concreteness effect.

p. 6 It would be helpful for the reader to briefly define motor imagery and to justify using motor imagery as opposed to action. Is the idea to show that this motor-to-semantic priming is robust and hence “works” with motor imagery and not just overt action?

Experiment 1

It is not clear why the authors hypothesize that participants in the KMI group would perform better in both tasks; this should be specified/made explicit. The same is true for the hypothesis concerning the abstract/action verb distinction.

Methods

In line 185 the authors wrote “We chose the KMI modality as it activates the motor neural network to a greater extent (42).” To a greater extent than what?

For experiment 2 it is noted that “Each training session lasted about 15 minutes” but no duration is given for experiment 1.

Again, for Experiment 2, it is hypothesized that participants would show greater improvement in processing action compared to abstract verbs. I’m assuming that the reason behind this is a belief that the motor cortex is more involved in representing abstract verbs but if this is the case, the story would be more coherent if it is stated in the general introduction or the Experiment introductions.

Discussion(s)

Line 385: “the results showed a significant main effect of the training group, suggesting that KMI training can facilitate lexical-semantic processing. This facilitation effect could be explained by a motor pre-activation induced by KMI but not by SVI.”

A semantic-to-motor priming effect, but also interference when motor preparation and compatible semantic processing occur simultaneously, has been largely discussed in the Action-Sentence compatibility effect literature. The authors might want to include a few of these references in their introduction/discussion (Aravena et al., 2012; Borreggine, 2008; Boulenger et al., 2006; de Vega et al., 2013; Diefenbach et al., 2013; Dudschig et al., 2014; Glenberg and Kaschak 2002; Zwaan & Taylor, 2006). In this same context, it could be interesting to discuss García and Ibáñez’s HANDLE model (2016).

Line 677 – “there are some factors not taken into consideration, such as gender and age” – are these important factors? Why? Predictions?

Line 687 - “Some previous studies found that motor-based training protocols can improve abstract verb processing” – At first this seems to contradict the statement on line 659 “… but to our knowledge, there are no studies that directly examined the effect of MI training on language comprehension tasks.” I think it might be helpful to point out that the motor-based training protocols referred to in line 687 involve actual action as opposed to MI.

737 – Another place to possibly mention the TMS study described above, under Introduction

Conclusion

761 – “it relies on the abilities of various patients’ – I’m not sure what this means, I think it requires clarification

Minor Comments

English

I would suggest that the writers have the manuscript proofread by a native English speaker as some of the wording is sometimes awkward, stilted or too informal.

Below is a short non-exhaustive list with examples

107 – “that had effects on language functions” � that affected language functions

121, 121 “In a first step” “In a second step” � “First,” “Following this,”. Step 1 and Step 2 can be added in parentheses, as they are for Step 3.

131 – “participants had to judge as fast as possible” � were asked to judge as quickly as possible.

145 – “To be included, they had to be right-handed, native French speakers ….” � Participants were right-handed, native French speakers.

152 – “After receiving an explanation of the procedures, participants…”

158 – “we told participants that they were…” – “participants were told…”

204/205 – “Participants had to imagine” � Participants were asked to imagine.

206 – “After reminding the instructions” �After being reminded of the instructions

218: “that are related to an action trained in the MI training phase…” � related to an action from (or presented during) the training phase

225-227 – “the” can be taken out (i.e., lexical frequency, number of homophones, number of letters…)

256 – “Following” � Following this

258 – “The pictures stayed on the screen” � The pictures remained on the screen

259 – “There were three categories of sentences” � The sentences fell under three categories

260 – “Sentences that contained one of the 14 target verbs”

261 – “one of the non-target verbs”

367 – “neither” � either

424 – “was supposed to” � aimed at further recruiting

139/426/623/643 etc. – “more important” � greater

433 – “To be included, they had to be” � Participants were right-handed, native French speakers.

457 – Participants took part in four training sessions: the first one on the same day they completed the pre-training assessment and on the following three days.

486 – “Wrong” � incorrect

550 – “a better” � greater

567 – take out “Results in terms of”

598 – “would improve” � improved

657 – Several studies have examined

663 – “The present study had several advantages”: In English, an advantage is generally in relation to something else so you could say “The present study had several advantages in relation to other studies in this area” and site these studies, if that’s the case.

669 – “The present study failed to include” � The present study did not include

677 – “there are some factors not taken into consideration” � other, important, factors were not taken into consideration

694 – “For the abstract verb” � For abstract verbs

701 – “Improved significantly more” � improved significantly

709 – take out “had”

716 – take out “some”

723 – “it would not be expected that modulations of the motor system” -� modulations in the motor system would not be expected to influence…

725 – “and it would not be expected for the KMI group to have performed …” � and the KMI group would not have been expected to perform better…

746 – can facilitate

748 – take out “A”

Line 214 - repetition of the word “presented”

Parts of the Methods are in the future

Line 235 - “the 42 experimental verbs were presented twice”

Line 362 – and a total of… (no comma)

Line 363 - In terms of accuracy, participants

It might be the version I received by all the figures are a bit blurry (especially the scale numbers). Also, for figure 2, the grey on “SVI” for “Sentence-picture matching” is not as dark as the grey on the figure.

597 – “The second experiment” � It

603 – concreteness. This is effect is marginally significant, indicating that� concreteness, indicating that

657 – This paragraph has several sentences starting with “The present study”, which is very repetitive

6. PLOS authors have the option to publish the peer review history of their article (what does this mean?). If published, this will include your full peer review and any attached files.

Reviewer #1: No

Reviewer #2: No

---

## [Author Response · Author response to Decision Letter 0]

12 May 2022

Dear Editor, and dear reviewers 

We are thankful for having the opportunity to submit a revised draft of our manuscript. We appreciate the time and effort that you and the reviewers have dedicated to assessing our manuscript and are grateful for the insightful comments and feedback you provided. We are very grateful with the reviewer for their detailed and meticulous review of our manuscript. We modified the manuscript according to their suggestions, and reviewed the manuscript at length at their suggestion. These changes are highlighted within the manuscript in red.

Please, find a point-by-point response to the reviewers’ comments below.

Sincerely, 

Marcela Perrone-Bertolotti for all of the authors

---

## [Decision Letter · Decision Letter 1]

9 Jun 2022

Kinesthetic Motor-Imagery Training Improves Performance on Lexical-Semantic Access

PONE-D-22-03552R1

Dear Dr. Perrone-Bertolotti,

We’re pleased to inform you that your manuscript has been judged scientifically suitable for publication and will be formally accepted for publication once it meets all outstanding technical requirements.

Kind regards,

Victor Frak, MD, Ph.D

Academic Editor

PLOS ONE

Additional Editor Comments (optional):

Dear Marcela,

There are still some small details in the grammar of the paper that will surely be corrected by you during the final publication process.

Thank you for submitting your interesting work to PLOS One.

Cordially

V Frak

Reviewers' comments:

Reviewer's Responses to Questions

**Comments to the Author**

1. If the authors have adequately addressed your comments raised in a previous round of review and you feel that this manuscript is now acceptable for publication, you may indicate that here to bypass the “Comments to the Author” section, enter your conflict of interest statement in the “Confidential to Editor” section, and submit your "Accept" recommendation.

Reviewer #1: (No Response)

Reviewer #2: All comments have been addressed

2. Is the manuscript technically sound, and do the data support the conclusions?

Reviewer #1: Yes

Reviewer #2: Yes

3. Has the statistical analysis been performed appropriately and rigorously? 

Reviewer #1: Yes

Reviewer #2: Yes

4. Have the authors made all data underlying the findings in their manuscript fully available?

Reviewer #1: Yes

Reviewer #2: Yes

5. Is the manuscript presented in an intelligible fashion and written in standard English?

Reviewer #1: Yes

Reviewer #2: Yes

6. Review Comments to the Author

Reviewer #1: I appreciate the work done by the authors to address my comments and I find that the manuscript significantly improved, for this reason, I endorse its publication.

Reviewer #2: This is a new version of a previous review article. The authors have answered all my question in a satisfactory manner. I find this version much improved, especially as regards the theoretical background, which is now much more complete. Also, the interpretation of the results in the discussion seems more thorough now, given the richer theoretical background to which it refers. I also found the English much improved. I only have a few very minor corrections.

Line 62. On the contrary  on the other hand

Line 85. For instances  for instance

Lines 209 and 210 “in one hand”, “in the other hand” use “on”

Rely in rely on

… if the training expand  if the training expands

Line 89. The authors discuss a TMS study and then use the term “neuroimagining” to describe these studies. I would chose a different term.

Line 95. And, more specifically, provided results …

Line. 201 is to be expected from participants in the KMI group compared to that of the SVI control group

Line 516. close parentheses

results on results from

Line 519. But is able to extend  but extends

Line 568. Will improve  would improve

7. PLOS authors have the option to publish the peer review history of their article (what does this mean?). If published, this will include your full peer review and any attached files.

Reviewer #1: No

Reviewer #2: No

---

## [Editor Report · Acceptance letter]

14 Jun 2022

PONE-D-22-03552R1 

Kinesthetic Motor-Imagery Training Improves Performance on Lexical-Semantic Access 

Dear Dr. Perrone-Bertolotti:

I'm pleased to inform you that your manuscript has been deemed suitable for publication in PLOS ONE. Congratulations! Your manuscript is now with our production department. 

Kind regards, 

on behalf of

Dr. Victor Frak 

Academic Editor

PLOS ONE